# Multiplex Assays in Allergy Diagnosis: Allergy Explorer 2 versus ImmunoCAP ISAC E112i

**DOI:** 10.3390/diagnostics14100976

**Published:** 2024-05-08

**Authors:** Hannes Nösslinger, Ewald Mair, Gertie J. Oostingh, Verena Ahlgrimm-Siess, Anna Ringauf, Roland Lang

**Affiliations:** 1Department of Nutrition and Dietetics, Hospital of Merano (SABES-ASDAA), Lehrkrankenhaus der Paracelsus Medizinischen Privatuniversität, 39012 Merano-Meran, Italy; 2Registered pediatrician (SABES-ASDAA), 39031 Brunico-Bruneck, Italy; ewald.mair@web.de; 3Biomedical Sciences, Salzburg University of Applied Sciences, 5431 Puch/Salzburg, Austria; geja.oostingh@fh-salzburg.ac.at; 4Department of Dermatology and Allergology, University Hospital of the Paracelsus Medical University, 5020 Salzburg, Austria; v.ahlgrimm-siess@salk.at (V.A.-S.); r.lang@salk.at (R.L.); 5MacroArray Diagnostics (MADx), 1230 Vienna, Austria; ringauf@macroarraydx.com

**Keywords:** ALEX^2^, allergies, ISAC, molecular diagnostics, multiplex assay

## Abstract

ImmunoCAP ISAC E112i (ISAC) and Allergy Explorer 2 (ALEX^2^) detect specific immunoglobulin E (IgE) reactivity. Both multiplex assays contain molecular allergens and ALEX^2^ additionally includes allergen extracts and inhibitors that block the binding of IgE to cross-reacting carbohydrate determinants (CCDs). This study aimed to compare the performance of ISAC and ALEX^2^ by determining the IgE reactivity against allergen extracts and/or allergen components and by using qualitative, semiquantitative, and quantitative analyses of all comparable allergen components in sera from 216 participants recruited in South Tyrol/Italy. For extract sensitization in ALEX^2^, the analysis revealed negative corresponding allergen components in 18.4% and at least one positive corresponding allergen component in 81.6% of all cases. For ISAC, the corresponding results were 23.5% and 76.5% of cases, respectively. The ALEX^2^ CCD inhibitor eliminated CCD-positive signals detected by ISAC in 88.5% of cases. Based on sensitization values of 0.3–14.9 ISU or kU_A_/L, there was good agreement between ALEX^2^ and ISAC, although ALEX^2^ showed higher values than ISAC. The addition of allergen-extract tests in ALEX^2^ resulted in the detection of more sensitizations than with corresponding allergen components alone. In the range of <15 ISU or kU_A_/L, ALEX^2^ may be more effective in detecting sensitizations.

## 1. Introduction

Molecular-based allergy diagnostics, also known as precision allergy molecular diagnostic applications (PAMD@s) [1], have opened up a new dimension in allergy diagnosis. Integrating novel individual molecular data enables precise diagnosis and therapy, leading to precision [2] and personalized medicine [3]. PAMD@s can be performed using singleplex, multi-allergen, or multiplex assays. Multi-allergen and multiplex assays simultaneously analyze approximately 10 and >100 allergens, respectively [4]. Although the sensitivity of singleplex assays is higher than that of multiplex assays, the latter is advantageous owing to its simultaneous analysis of immunoglobulin E (IgE) against multiple allergens in a single blood sample [1,4]. Therefore, multiplex assays are useful in polysensitized patients or in cohort studies to track molecular spreading, which is the increase in sensitizations over time [5]. Recommended procedures for allergy diagnosis start with a clinical history, followed by the skin prick test (SPT), extract diagnosis, and ultimately singleplex or multiplex PAMD@ [4]. In addition, in some cases, the new concept of “from molecules to clinic” can be useful, e.g., to lower skin test numbers in young children, in polysensitized atopic patients, or whenever skin tests become difficult or unreliable [6]. Multiplex assays have become indispensable tools in allergy diagnostics and are increasingly used in daily routines.

The widely used multiplex assay ImmunoCAP ISAC E112i (ISAC) and a newer multiplex assay Allergy Explorer 2 (ALEX^2^) can analyze IgE reactivity against 112 allergens from 51 allergen sources and 295 allergens from 165 allergen sources, respectively, in a single blood sample [7,8]. The ALEX^2^ microchip is novel because it comprises 117 allergen extracts in addition to 178 molecular allergen components and provides inhibitors that suppress the binding of IgE to clinically irrelevant, cross-reacting carbohydrate determinants (CCDs) [8].

Comparative studies on ISAC and ALEX^2^ have been reported [9,10,11,12]. However, in contrast to previously published studies, we aimed to comprehensively investigate the additional value of the ALEX^2^ allergen extracts, the increased number of available allergen components, and the impact of included CCD inhibitors by comparing the ISAC and ALEX^2^ tests using qualitative, semiquantitative, and quantitative analytical methods. Furthermore, we are interested in the comparison of the two multiplex assays’ performance.

## 2. Materials and Methods

### 2.1. Study Design and Ethical Approval

This study was a prospective, observational, descriptive, monocentric, and cross-sectional study (level 4).

Procedures adhered to tenets of the Declaration of Helsinki, and the study was approved by the Ethics Committee of the South Tyrolean Health Authority (approval number: 135-2020). All participants signed an informed consent form.

### 2.2. Recruitment and Examination Procedure

The study participants were recruited in South Tyrol, an alpine region in the north-east of Italy without access to the sea. Inclusion criteria were the presence of allergic symptoms and suspected allergies, as well as positive sensitization to aeroallergens or food allergens that we or other physicians were able to detect with sIgE measurements or skin prick tests during previous patient examinations. The only exclusion criteria in reference to medical history was pregnancy to avoid any potential harm.

The examinations of the participants were performed in Brunico, South Tyrol, Italy. Each of the 216 participants was assessed at least once. The examination period, including blood extraction for serum collection and the simultaneously performed ISAC and ALEX^2^ tests, took place from 1 February 2021 to 5 November 2021. Participants’ allergy history was obtained at the beginning of each examination. An allergy history questionnaire (QUETHEB allergy questionnaire from the German Society of Qualified Nutritional Therapists and Nutritionists) [13] was used to aid in standardizing symptom description.

Venous blood was collected (Greiner Bio-One S.r.l Vacuette^®^ Tube 7 mL CAT Serum Separator Clot Activator, Cassina de Pecchi, Italy) from fossa cubiti or dorsum manus of each participant and left undisturbed for 15–30 min to allow blood clotting. Thereafter, the serum was obtained after centrifugation at 1500× *g* for 10 min in a refrigerated centrifuge, and 2 mL serum was transferred into each of 2 new tubes and stored at −20 °C (EVERmed LF Laboratory Freezer, Motteggiana, Italy). These frozen samples were transported to the 2 laboratories. Frozen sera were transported in a portable EVERmed refrigerator PR11 (EVERmed, Motteggiana, Italy) at −18 °C. The ISAC and ALEX^2^ tests were performed at the Clinical Laboratory of Merano Franz Tappeiner Hospital, Merano, and the MacroArray Diagnostics (MADx), Vienna, respectively, according to the manufacturer’s instructions. Each blood sample tube and ISAC or ALEX² microchip was carefully checked visually before measurement to exclude potential disturbing factors.

### 2.3. Multiplex Platforms, ISAC and ALEX^2^

The ISAC and ALEX^2^ platforms detect specific immunoglobulin E (IgE) reactivity against purified natural and recombinant allergens or, in the case of ALEX^2^, also detect allergen extracts, immobilized on microchips. ISAC and ALEX^2^ testing was performed according to the manufacturer’s instructions.

ISAC test results (Thermo Fisher Scientific, Uppsala, Sweden) were reported in ISAC standardized units for IgE (ISU) and categorized into four specific IgE classes in the 0.3–100 ISU range (semiquantitative): negative (<0.3 ISU), low (0.3–0.9 ISU), moderate to high (1–14.9 ISU), and very high (≥15 ISU).

ALEX^2^ test results (MacroArray Diagnostics, Vienna, Austria) were reported in kilounits of allergen-specific IgE per litre (kU_A_/L) and categorized into five specific IgE classes in the 0.3–50 kU_A_/L range (quantitative): negative or uncertain (<0.3 kU_A_/L), low (0.3 to <1 kU_A_/L), moderate (1 to <5 kU_A_/L), high (5 to <15 kU_A_/L), and very high (≥15 kU_A_/L). The ALEX^2^ test’s range of total IgE was 1–2500 kU_A_/L (semiquantitative).

### 2.4. Statistics

To compare the two allergy multiplex platforms, ISAC and ALEX^2^, 216 serum samples from patients with known or suspected allergies were analyzed. The sample size was calculated using Bland–Altman plots of preliminary studies as conducted by Lu et al. [14]. The type I error (α) was set at 0.05, the power (1–β) at 0.80, and the confidence interval at 95%. The residuals were tested for normal distribution using the Pearson chi-square normality test. Quantitative, semiquantitative, and qualitative comparative analyses were performed. Furthermore, we analyzed whether the additional testing of extracts leads to the detection of more sensitization and to what extent CCD inhibition influences sensitization.

#### 2.4.1. Qualitative Comparison

For qualitative comparison, first, the dichotomized ISAC and ALEX^2^ values (negative, <0.3 ISU or kU_A_/L; positive, ≥0.3 ISU or kU_A_/L) were examined via a qualitative inter-assay comparison (weighted Cohen’s kappa), and the Overall Rates of Agreement (ORAs), positive and Negative Percent Agreement (PPA and NPA), and concordance percentage with discarded double-negative (DND) and double-positive data (DPD) were calculated. Kappa values were interpreted as poor (<0.20), fair (0.21–0.40), moderate (0.41–0.60), good (0.61–0.80), or very good (0.81–1.00) [15].

#### 2.4.2. Semiquantitative Comparison

For the semiquantitative study, ISAC and ALEX^2^ values were grouped (class 0: <0.3 ISU or kU_A_/L; class 1: 0.3 to <1.0 ISU or kU_A_/L; class 2: 1 to <5.0 ISU or kU_A_/L; class 3: 5 to <15 ISU or kU_A_/L; and class 4: ≥15 ISU or kU_A_/L), and the weighted Cohen’s kappa and class agreement rates were calculated.

#### 2.4.3. Quantitative Comparison

For quantitative comparison, Spearman’s [16] and Lin’s [17] correlation coefficients were calculated. Lin’s coefficient values of <0.90, 0.90–0.95, 0.95–0,99, and >0.99 were indicative of poor, moderate, good, and excellent reliability, respectively.

The calculated intraclass correlation coefficient (ICC) was interpreted as follows: <0.5, 0.5–0.75, 0.75–0.9, and >0.9 were indicative of poor, moderate, good, and excellent reliabilities, respectively.

The Bland–Altman analysis was used for method comparison, and the results were presented graphically in the form of Bland–Altman and Residual plots together with Mountain plots. IgE cross-reactivity in ISAC and ALEX^2^ was compared between different pan-allergen groups using the Spearman correlation coefficient. ALEX^2^ allergen extracts were compared with the corresponding components in ISAC and ALEX^2^ using correlation analysis and the influence of ALEX^2^-CCD inhibition on the specific IgE of Cyn d 1 (Bermuda grass), Cry j 1 (Japanese cedar), and Cup a 1 (Arizona cypress) was investigated.

Statistical analysis was performed using the fee-based MedCalc^®^ Statistical Software version 20.110 (MedCalc Software Ltd., Ostend, Belgium; https://www.medcalc.org; accessed on 10 June 2020) and R version 3.5.1 ((7 February 2018) 2018 The R Foundation for Statistical Computing and RStudio, version 1.2.1335, 2009-2019 R Studio, Inc., Boston, MA, USA).

## 3. Results

### 3.1. Allergic Symptoms

In total, 216 participants, including children and adults, were included (93 [43%] females and 123 [57%] males) with a median age of 24 years (1–76 years). Seasonal symptoms were present in 127 (59%) patients, perennial symptoms in 28 (13%), and perennial symptoms with seasonal deterioration in 61 (28%). Based on the subjective assessment results, the symptom intensity was distributed as follows: 19 (9%) low; 19 (9%) low to medium; 90 (42%) medium; 24 (11%) medium to high; and 64 (29%) high. Respiratory, gastrointestinal, and skin symptoms were reported by 205 (95%), 82 (38%), and 53 (25%) patients, respectively. Anaphylaxis was experienced by 17 (8%) patients at least once. Moreover, 18 (8%) patients had completed subcutaneous allergen immunotherapy and 8 (4%) patients had completed sublingual allergen immunotherapy in the past.

### 3.2. Extracts

ALEX^2^ test allergen extracts provided additional information regarding sensitization, although this did not apply equally to all allergen extracts.

Ten extracts without corresponding components in ALEX^2^ showed a prevalence rate of >5% in our collective: Hel a (sunflower seed) 5.1%; Ach d (house cricket) 5.6%; Ama r (pigweed) 6.0%; Ten m (mealworm) 6.5%; Can f male urine (dog male urine) 7.4%; Car i (pecan) 7.45%; Phr c (common reed) 10.2%; Jug r pollen (walnut pollen) 15.3%; Pas n (bahia grass) 43.1%; and Sec c pollen (rye pollen) 66.7%.

Even in cases where corresponding components are available for extracts, the components cannot always replace the extracts. The example of Salsola kali (Sal k) shows that, in our collective, 94.7% sensitization can be detected with the extract Sal k, which would have remained undetected with the component Sal k 1 (Appendix A).

### 3.3. CCD

MUXF3 was used as a CCD marker in the ISAC test and was positive in 24.1% (52/216) of all cases. The ALEX^2^ assay’s CCD inhibitor decreased the number of cases with CCD-positive signals compared to the ISAC test by 88.5% (46/52 of samples). However, IgE binding to CCDs was not inhibited in 6 out of 52 patients, who tested positive for Hom s LF, the CCD marker used in the ALEX^2^ test (Appendix A Appendix A).

In the ISAC test, four native aeroallergen components are common causes of CCD reactivity (Cup a 1, Cry j 1, Cyn d 1, and Phl p 4 (timothy)). Three of these could be directly compared between ISAC and ALEX^2^ (Cup a 1, Cry j 1, and Cyn d 1). At <15 ISU or kU_A_/L, the ISAC test detected more sensitizations than ALEX^2^ for Cup a 1, Cry j 1, and Cyn d 1, which may be due to the successful inhibition of CCDs in the ALEX^2^ assay (Table 1).

A greater number of higher ALEX^2^ values were measured within class 4 (≥15 ISU and kU_A_/L) compared with ISAC. Thus, CCD inhibition in the high-value ranges (≥15 ISU or kU_A_/L) may be less effective when compared to the inhibition in the lower ranges (<15 ISU or kU_A_/L).

### 3.4. Sensitizations, Allergy Symptoms, and Pollen/Spore Calendar

Correlations between sensitization, allergy symptoms, and the pollen/spore calendar differed more by pollen type than by test type. The match of allergy symptoms with the pollen/spore calendar when ALEX^2^ was used to analyze sensitization to Alternaria, grass pollen, tree pollen, and weed pollen was 81.8%, 76.7%, 50.6%, and 41.0%, respectively. When ISAC was used to analyze sensitization to Alternaria, grass pollen, tree pollen, and weed pollen, the match of allergy symptoms with the pollen/spore calendar was 80.0%, 77.9%, 55.0%, and 38.4%, respectively (Appendix A Appendix A).

### 3.5. Qualitative, Semiquantitative, and Quantitative Comparison

ISAC and ALEX^2^ are semiquantitative and quantitative tests, respectively. Both tests have different higher-value ranges; hence, purely quantitative data comparison is not very helpful. For this reason, a qualitative and semiquantitative comparison of methods was unavoidable despite the lower amount of data. For the sake of completeness, we have not excluded the quantitative comparison.

#### 3.5.1. Qualitative Comparison

Very good agreement was found between ISAC and ALEX^2^ regarding dichotomous variables (negative and positive), with an overall kappa value of 0.86. The concordant agreement between both tests was 97.3%, which was reduced to 77.3% when double-negative results were discarded (Table 2).

When comparing the ability of ISAC and ALEX^2^ to identify pan-allergens, differences were found regarding individual pan-allergen groups, e.g., for profilins (21.8% in ISAC vs. 15.6% in ALEX^2^) and for PR-10 proteins (31.9% in ISAC vs. 27.9% in ALEX^2^), which were more frequently positive in ISAC (Table 3).

More heterogeneous results emerged when comparing the overlap of positively tested allergens within individual pan-allergen groups with ISAC and ALEX^2^ (e.g., co-recognition ranged from 60.7% for profilin to 84.0% for lipocalin) (Figure 1).

#### 3.5.2. Semiquantitative Comparison

For the semiquantitative comparison between ALEX^2^ and ISAC, a class agreement was calculated. The highest agreement between ISAC and ALEX^2^ was found in classes 0 and 4 (98.9% and 87.8% for all comparable allergens, respectively), whereas class 1 always scored the lowest (22.0%) because of the small range and low number of comparable values in both ISAC and ALEX^2^.

Inhalant allergens (kappa value of 0.86) showed higher agreement than food allergens (kappa value of 0.74) regarding class and qualitative (negative and positive) comparisons (Figure 2).

Within classes, different allergen groups performed differently, with molds and yeasts showing the best performance for classes 0, 2, and 4; grass pollen for class 1; and tree pollen for class 3.

#### 3.5.3. Quantitative Comparison

Lin’s Concordance Correlation Coefficients (CCCs) indicated the poor strength of the agreement for all comparable (0.83), inhalative (0.83), and food (0.77) allergens. Therefore, differing dynamic ranges, particularly at high values, must be considered when comparing the raw data of ISAC and ALEX^2^. This was also expressed in the regression equation (y = 0.0 + 1.63x [95% CI, 1.55–1.72]) with its proportional error. Residuals between sIgE in ISAC and ALEX^2^ were normally distributed (*p* < 2.2 × 10^−16^).

However, the intraclass correlation coefficients (ICCs, average measures for allergen groups) were >0.75, except for Ani s (Anisakis simplex) and Hev b (latex), indicating good agreement between ISAC and ALEX^2^.

In the Bland–Altman plot, there was only a small constant bias (−0.24 ISU or kU_A_/L for the sum of all comparable allergens, and for individual allergens ranging from a minimum of 0 to a maximum of −2.64 ISU or kU_A_/L for Der f 2 (American house dust mite)), and narrow lines of agreement (LoA) in the lower range (good agreement), but much broader LoA in the upper range (poorer agreement). High-grade sensitizations, which typically occur in inhalant or pollen allergies, corresponded with widely spaced LoA, in contrast to food allergies. In the lower value ranges (classes 1–3 or <15 ISU or kU_A_/L), trends were found with negative slopes and higher ALEX^2^ values compared to ISAC, whereas in class 4 (≥15 ISU or kU_A_/L), an overall positive slope with a trend reversal point at 40 ISU or kU_A_/L was observed (Figure 3).

## 4. Discussion

The ALEX^2^ multiplex assay provides spotted allergen extracts in addition to molecular allergen components. The results of this study showed that the allergen extracts and a higher number of allergen components in ALEX^2^ provided an extensive range in the number of positive results (ISAC 2655 vs. ALEX^2^ 4101), although comparable components of ISAC (2330) showed a slightly higher number of sensitizations than ALEX^2^ (2180). Sensitizations to allergen extracts that could not be identified by the corresponding allergen components, and vice versa, have already been reported by Scala et al. [9] and Quan et al. [12]. Data from our study further demonstrated that testing allergen extracts can be useful to complement the detection of sensitizations to allergens from various allergen sources, e.g., cow’s milk or hen’s egg, where the sensitivity of the molecular components of both platforms is low (Appendix A).

Comparing the two assays, ISAC and ALEX^2^ antibodies against CCD epitopes did not play a major role in our study. In ALEX^2^, 86 allergen extracts and components contain CCD epitopes. We compared the commonly CCD-reactive components Cup a 1, Cry j 1, and Cyn d 1 in ISAC with the corresponding sensitizations in ALEX^2^ (Phl p 4 is unavailable in ALEX^2^). The results showed that CCD inhibition is best when the CCD IgE value is <15 ISU or kU_A_ /L, which was contrary to the findings of Scala et al. [9] suggesting that CCD inhibition is independent of the level of CCD IgE signals. However, they considered the level and inhibition of MUXF3 and Hom s LF signals rather than a direct comparison of allergen components. As we evaluated only three comparable commonly CCD-reactive components in both tests, this assumption cannot be extrapolated to all CCD-reactive components in ALEX^2^.

To the best of our knowledge, the correlation in this study between sensitization and inhalative allergens measured using ISAC and ALEX^2^, allergy symptoms, and the pollen/spore calendar is a completely new approach. The pollen/spore calendar of the examination year and pollen traps in the vicinity of the study participants’ homes were used for the evaluation. Nevertheless, some uncertainties remain in attributing allergic symptoms to the triggering allergen source due to polysensitization. The highest correlation was found between allergic symptoms, the pollen/spore calendar, and sensitization to grass pollen and Alternaria, which often presents as co-sensitization with pollen. However, the correlation between ISAC and ALEX^2^ was less accurate for tree and weed pollen. Nevertheless, this study’s findings are likely to have practical significance, as both ISAC and ALEX^2^ have shown similar results in pollen allergy diagnosis.

A direct comparison between the ISAC and ALEX^2^ test results is hindered by different testing methods (semiquantitative for ISAC and quantitative for ALEX^2^) and different dynamic value ranges (especially at higher specific IgE levels). Therefore, the qualitative comparison of platforms is probably more relevant than the quantitative comparison.

In our study, ISAC and ALEX^2^ values showed very good agreement based on the quantitative comparison of their dichotomized values, with a better total kappa value of 0.857 than that reported by Scala et al. (0.795) [9], an excellent Negative Percent Agreement (NPA) of 98.9%, and a Positive Percent Agreement (PPA) of 84.4%. The overall rate of agreement (ORA) was 97.3%, which reduced to 77.3% when double-negative results were discarded (DND). Dichotomizing positive and negative results is important for screening and excluding sensitization and suspected allergies. This indicates that a good exclusion of sensitization was achieved by screening. There were differences between the various allergen groups in our study; however, the deviations were small. The analysis of the 102 comparable allergens resulted in a very good, good, moderate, fair, and poor agreement for 55 (53.9%), 20 (19.6%), 11 (10.8%), 3 (2.9%), and 13 (12.8%) allergens, respectively. Similar to the findings of Nerelius et al. [18], the comparison of ALEX^2^ with ISAC resulted in 15.5% false-negative and 1.1% false-positive results for ALEX^2^.

In the semiquantitative comparison between ALEX^2^ and ISAC, we found the best agreement for the classes with < 0.3 ISU or kU_A_/L, at 98.9%, and with >15 ISU or kU_A_/L, at 87.8%, which was similar to findings of Nerelius et al. [18], and the worst agreement for the class was with 0.3–1.0 ISU or kU_A_/L, at 22%, due to the narrow class range. Food allergens, weed pollen, Anisakis, and Hevea also performed worse than the other allergen groups in the class comparison.

For a quantitative comparison, contradictory results have been reported in the literature [9,19,20], ranging from good agreement to no concordance. In our Bland–Altman plots, only a small constant bias (systematic error of −0.24 ISU or kUA/L), narrow LoA in the lower ranges (good agreement), and broad LoA in the upper range (poor agreement) were observed. Contrary to previously published studies, we focused on the entire measurement range rather than individually measured values and performed Bland–Altman plots for different measurement ranges. The plots for classes with 0.3–0.9 ISU or kU_A_/L (class 1), 1.0–4.9 ISU or kU_A_/L (class 2), and 5.0–14.9 ISU or kU_A_/L (class 3) showed mostly higher ALEX^2^ values than ISAC values with a negatively sloped trend line. However, at 40 ISU or kU_A_/L (class 4), higher ISAC values were observed owing to the different dynamic ranges of the two assays. We conclude from our observation that at <15 ISU or kU_A_/L, ALEX^2^ may be better suited to detect sensitization in the screening process due to its higher values compared to ISAC.

A limitation of this study is the small number (n = 216) of participants. Nonetheless, to date, our study is the first to compare ISAC and ALEX^2^ using >200 samples. Furthermore, the study design could be improved by a comparison of the two methods with a gold standard, which would be a double-blind placebo-controlled food challenge for food allergies. However, for inhalant allergens, there is no gold standard [4]. In addition, the evaluation of ISAC and ALEX^2^ was performed in two different laboratories by professional technical assistants, which could have introduced bias. On the one hand, we should also take into account, when comparing two different IgE detection technologies, that the results obtained should not be treated as identical. Inconsistent results between different multiplex assays may be related to differences in the adopted reference standard, the method of obtaining the calibration curve, and the allergens used in the method. On the other hand, ISAC and ALEX^2^ are competing multiplex assays. In everyday practice, physicians usually have to decide in favour of one or the other multiplex test and want to know how they perform differently.

## 5. Conclusions

Allergen extracts in ALEX^2^ can aid the detection of more sensitizations than the corresponding allergen components alone, but cannot replace them. Furthermore, the correlation between sensitization, allergy symptoms, and the pollen/spore calendar differed more by the pollen than by the test types used to measure sensitization. Therefore, both tests’ results are comparable. Finally, ALEX^2^ may be better suited to detect sensitization at <15 ISU or kU_A_/L in the screening process.

## Figures and Tables

**Figure 1 diagnostics-14-00976-f001:**
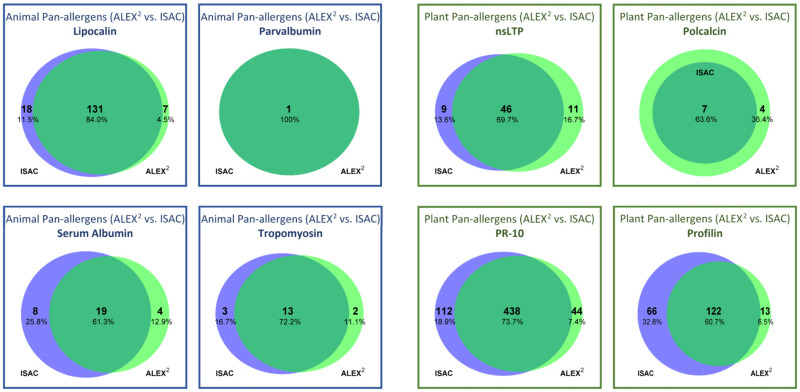
Frequency of detection of animal and plant pan-allergens by ISAC (blue) and ALEX^2^ (green) and their concordance.

**Figure 2 diagnostics-14-00976-f002:**
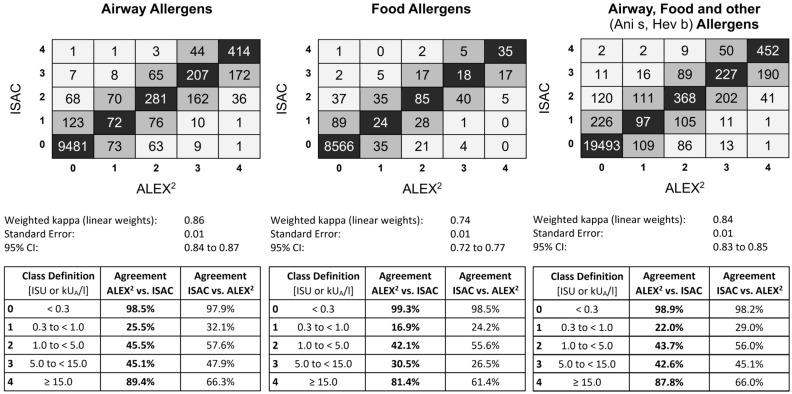
Heatmap and class agreement between ISAC and ALEX^2^.

**Figure 3 diagnostics-14-00976-f003:**
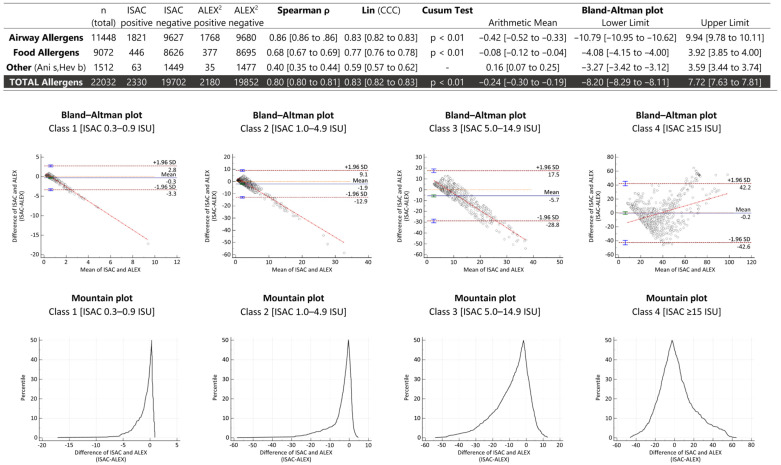
Bland–Altman and Mountain plots of quantitative class comparison of all allergen components.

**Table 1 diagnostics-14-00976-t001:** Class agreement between ISAC and ALEX^2^ for CCD-inhibited Cup a 1, Cry j 1, and Cyn d 1.

ISAC class 4	≥15 ISU	0.0	0.0	7.7	40.0	65.0
ISAC class 3	5 to <15 ISU	1.1	0.0	30.8	53.3	20.0
ISAC class 2	1 to <5 ISU	13.8	33.3	38.5	6.7	15.0
ISAC class 1	0.3 to <1 ISU	18.4	66.7	15.4	0.0	0.0
ISAC class 0	<0.3 ISU	66.7	0.0	7.7	0.0	0.0
Column Percentages		100%	100%	100%	100%	100%
		ALEX^2^ class 0	ALEX^2^ class 1	ALEX^2^ class 2	ALEX^2^ class 3	ALEX^2^ class 4
		<0.3 kU_A_/L	0.3 to <1 kU_A_/L	1 to <5 kU_A_/L	5 to <15 kU_A_/L	≥15 kU_A_/L

Class agreement for the comparable CCD reactive components Cup a 1, Cry j 1, and Cyn d 1 (MUXF3-positive in ISAC and Hom s LF-negative in ALEX^2^). The classification was adopted from ISAC and ALEX^2^. CCD, cross-reacting carbohydrate determinant; Cup a 1, Cupressus arizonica 1; Cry j 1, Cryptomeria japonica 1; Cyn d 1, Cynodon dactylon 1; MUXF3, CCD composed of N-acetyl glucosamine, mannose, fucose, and xylose; Hom s LF, Hom s Lactoferrin; ISU, ISAC standardized units; kU_A_/L, kilounits of allergen-specific IgE per litre.

**Table 2 diagnostics-14-00976-t002:** Overview of the qualitative evaluation values (total participants: 216). Calculations related to the dichotomization of ISAC and ALEX^2^ values.

	No. ^a^	ISAC Positive ^b^	ALEX^2^ Positive ^b^	ISAC Negative ^c^	ALEX^2^ Negative ^c^	Kappa ^d^	ORA ^d^ [%]	PPA ^d^ [%]	NPA ^d^ [%]	DND ^d^ [%]	DPD ^d^ [%]
Airway Allergens	11448	1821	1768	9627	9680	0.88	96.8	88.8	98.4	82.0	96.4
Pollen	5400	1337	1273	4063	4127	0.87	95.4	88.3	97.7	82.5	94.1
Grass Pollen	1512	759	776	753	736	0.91	95.4	96.6	94.3	91.4	91.1
Tree Pollen	2160	444	417	1716	1743	0.83	94.6	83.8	97.4	76.1	93.5
Weed Pollen	1728	134	80	1594	1648	0.68	96.3	56.0	99.7	54.0	96.1
Molds and Yeasts	1296	31	24	1265	1272	0.83	99.3	74.2	99.9	71.9	99.3
Furry Animals	2376	255	254	2121	2122	0.89	97.9	90.2	98.9	82.4	97.7
Mites and Cockroaches	2376	198	217	2178	2159	0.88	98.0	92.9	98.5	79.7	97.9
Food Allergens	9072	446	377	8626	8695	0.76	97.9	71.1	99.3	62.6	97.8
Other (Ani s, Hev b)	1512	63	35	1449	1477	0.64	97.8	50.8	99.8	48.5	97.7
TOTAL Allergens	22032	2330	2180	19702	19852	0.86	97.3	84.4	98.9	77.3	97.1

^a^ Number of components compared; ^b^ positive: ≥0.3 ISU or kU_A_/L; ^c^ negative: <0.3 ISU or kU_A_/L; ^d^ Weighted kappa: linear weights; ORAs, Overall Rates of Agreement; PPA, Positive Percent Agreement; NPA, Negative Percent Agreement; DND, Concordant Results with double negatives discarded; DPD, Concordant Results with double positives discarded; Ani s, Anisakis simplex; Hev b, Hevea brasiliensis.

**Table 3 diagnostics-14-00976-t003:** Comparison of pan-allergen groups.

Component Family	No. ^a^	ISAC Positive ^b^	ALEX^2^ Positive ^b^	ISAC [ISU] Mean Positive ^b^	ALEX^2^ [kU_A_/L] Mean Positive ^b^	Kappa ^c^	ORA ^c^ [%]	PPA ^c^ [%]	NPA ^c^ [%]	DND ^c^ [%]	DPD ^c^ [%]
2S Albumin	1728	36 (2.1%)	44 (2.5%)	16. 7 [8.6–24.8]	22.4 [14.9–30.0]	0.82	99.2	91.7	99.3	70.2	99.2
Serum Albumin	1080	27 (2.5%)	23 (2.1%)	10.8 [5.2–16.5]	15.5 [5.2–16.5]	0.75	98.9	70.4	99.6	61.3	98.9
7/8S and 11S Globulin	1296	24 (1.9%)	31 (2.4%)	5.8 [1.8–9.7]	12.0 [5.4–18.6]	0.76	99.0	87.5	99.2	61.8	99.0
Cystein Protease	648	79 (12.2%)	72 (11.1%)	9.8 [7.1–12.4]	9.5 [7.1–11.9]	0.89	97.7	86.1	99.3	81.9	97.4
Lipocalin	1512	149 (9.9%)	138 (9.1%)	9.5 [7.4–11.5]	13.0 [10.4–15.6]	0.90	98.3	87.9	99.5	84.0	98.2
NPC2-Family	648	94 (14.5%)	113 (17.4%)	14.1 [11.0–17.2]	21.3 [17.6–25.1]	0.85	95.8	95.7	95.8	76.9	95.2
nsLTP	1728	55 (3.2%)	57 (3.3%)	1.6 [1.1–2.1]	3.9 [2.6–5.1]	0.82	98.8	83.6	99.3	69.7	98.8
Polcalcin	216	7 (3.2%)	11 (5.1%)	8.3 [−1.0–17.5]	9.7 [0.3–19.0]	0.77	98.1	57.1	94.9	63.6	94.9
Profilin	864	188 (21.8%)	135 (15.6%)	5.2 [4.1–6.4]	4.8 [3. 5–6.2]	0.70	90.9	64.9	98.1	60.7	89.4
PR-10	1728	551 (31.9%)	482 (27.9%)	6.2 [5.3–7.2]	8.6 [7.5–9.7]	0.78	90.9	79.5	96.3	73.6	87.8
Tropomyosin	648	16 (2.5%)	15 (2.3%)	6.7 [2.5–10.9]	12.6 [5.2–20.0]	0.83	99.2	81.3	99.7	72.2	99.2

^a^ Number of components compared; ^b^ positive: ≥0.3 ISU or kU_A_/L; ^c^ Weighted kappa: linear weights; ORAs, Overall Rates of Agreement; PPA, Positive Percent Agreement; NPA, Negative Percent Agreement; DND, Concordant Results with double negatives discarded; DPD, Concordant Results with double positives discarded; nsLTP, non-specific lipid transfer proteins; PR-10, pathogenesis-related class 10 proteins; ISU, ISAC standardized units; kU_A_/L, kilounits of allergen-specific IgE per litre.

## Data Availability

The raw data supporting the conclusions of this article will be made available by the authors on request.

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
