# Peer review of "Multiplex Assays in Allergy Diagnosis: Allergy Explorer 2 versus ImmunoCAP ISAC E112i"

_diagnostics, 2024, doi:10.3390/diagnostics14100976_

Round 1
Reviewer 1 Report
Comments and Suggestions for Authors
This is a well described retrospective study comparing the performance of two assays using 216 patients. Extracts appears to provide more information as expected. The major limitation of the stud, which the author acknowledge, is lack of comparison with a reference method.
It is not clear of the authors in practice classify results as week, strong, etc.. particularly for those samples about the cut-off values. This is relevant given the degree of discordance reported by the authors
Author Response
April 29th 2024
Dear Reviewer,
We particularly thank you for your time and appreciate your valuable comment to improve the quality of the manuscript. Please find enclosed the answer to your comment. We hope we have addressed your raised point adequately.
“It is not clear of the authors in practice classify results as week, strong, etc.. particularly for those samples about the cut-off values. This is relevant given the degree of discordance reported by the authors”
The answer to this question cannot be generalized, as it depends on whether the comparison method is considered qualitative, semi-quantitative or quantitative.
In the qualitative comparison, positive and negative results with a cut-off value of 0.3 ISU or kUA/l are compared. In this respect, we report very good agreement with an overall kappa value of 0.86 (line 228-231).
In the semi-qualitative comparison, it is more difficult to classify the strength of agreement, as we examined 4 different sensitization classes for this comparison in addition to the kappa value. Whilst the agreement for inhalation allergens is very good with a kappa value of 0.86, it is good for food allergens with 0.74, and very good for all allergens together with a kappa value of 0.84. The class comparison shows very good agreement for negative and very high sensitization (class 0 and 4), the lowest for class 1 (0.3 to <1 ISU or kUA/l) as shown in Fig. 2 (lines 259-264).
In the quantitative comparison, we did not want to use classification such as weak, moderate, strong, or very strong, as the two test methods do not use the same units and ISAC is described as a semi-quantitative method and ALEX2 as a quantitative method. Therefore, reliable classification of the agreement between the two methods would be questionable.
Reviewer 2 Report
Comments and Suggestions for Authors
The manuscript entitled "Multiplex assays in allergy diagnosis: Allergy Explorer 2 versus ImmunoCAP ISAC E112i" compares the performance of both immunoassays in detecting IgE against allergens in a cohort of 216 patients at quantitative, semi-quantitative, and qualitative levels, while also correlating these findings with the clinical manifestations of the patients. The study design is robust, thus rendering the presented results both valid and insightful for interpretation. Nevertheless, there are several minor points to address to enhance the manuscript's overall quality.
Minor comments:
In line 85, it is stated that the patient inclusion period was February-November 2021. It is not clear when blood extraction for serum collection occurred, nor when the analyses with ALEX2 and ISAC were performed, whether simultaneously or sequentially.
Line 162. Using "patients" as the title for results seems awkward in reading; I believe "allergic symptoms" or "allergic clinical presentation" would be more fitting.
Line 220 states that quantitative method comparison is not useful, but then a quantitative analysis is performed. This contradiction is not explained.
Line 240. The difference in mean units between the two systems not being equivalent is not indicative of anything; therefore, it is understood that this result is not verifiable.
Figures 1, 2, and 3 are not included in the text and cannot be evaluated.
Line 267. I have serious doubts about the relevance of this analysis due to the lack of correlation between the units of the two techniques
Author Response
April 29th 2024
Dear Reviewer,
We particularly thank you for your time and appreciate your valuable comments and suggestions to improve the quality of the manuscript.
Please find enclosed the point-by-point answers to your comments including detailed descriptions of the revisions made to the manuscript. We hope we have addressed all the raised points adequately.
“In line 85, it is stated that the patient inclusion period was February-November 2021. It is not clear when blood extraction for serum collection occurred, nor when the analyses with ALEX2 and ISAC were performed, whether simultaneously or sequentially.”
Thank you for pointing out that we had not included this information. We have now added this information to the manuscript. “The examination period, including blood extraction for serum collection and the simultaneously performed ISAC and ALEX2 tests, took place from February 1st 2021, to November 5th 2021.”
“Line 162. Using "patients" as the title for results seems awkward in reading; I believe "allergic symptoms" or "allergic clinical presentation" would be more fitting.”
Thank you for the suggestion. We have changed the title to “Allergic Symptoms”.
“Line 220 states that quantitative method comparison is not useful, but then a quantitative analysis is performed. This contradiction is not explained.”
In line 220, we mentioned that a purely quantitative comparison is not useful. We changed the text to: “hence, purely quantitative data comparison is not very helpful. For this reason, a qualitative and semi-quantitative comparison of methods is unavoidable despite the lower level of data. For the sake of completeness, we have not excluded the quantitative comparison.”
“Line 240. The difference in mean units between the two systems not being equivalent is not indicative of anything; therefore, it is understood that this result is not verifiable.”
Thank you for pointing this out. You are absolutely right, which is why we have deleted this sentence.
“Figures 1, 2, and 3 are not included in the text and cannot be evaluated.”
We are sorry, we cannot understand why the figures are not visible. Before submission, we contacted the Managing Editor of the Diagnostics Editorial Office to make sure that we had included tables and figures correctly in the manuscript. We received confirmation from Managing Editor, Mr. Dennis Zhu, Diagnostics Editorial Office that everything was correct and that we could submit the manuscript in this form.
Anyway, this time we have inserted the images in the manuscript in png format and not in pdf format.
“Line 267. I have serious doubts about the relevance of this analysis due to the lack of correlation between the units of the two techniques.”
According to your comment to line 220, we relativized the value of the quantitative analysis. However, since quantitative comparisons between ISAC and ALEX2 are repeatedly made in the literature, e.g. Scala et al. 2021, we would also like to present the quantitative analysis in our paper.